# Activity of Amphotericin B-Loaded Chitosan Nanoparticles against Experimental Cutaneous Leishmaniasis

**DOI:** 10.3390/molecules25174002

**Published:** 2020-09-02

**Authors:** Alaa Riezk, Katrien Van Bocxlaer, Vanessa Yardley, Sudaxshina Murdan, Simon L. Croft

**Affiliations:** 1Department of Infection Biology, London School of Hygiene & Tropical Medicine, London WC1E 7HT, UK or alaa.riezk@lshtm.ac.uk (A.R.); katrien.vanbocxlaer@york.ac.uk (K.V.B.); vanessa.yardley@lshtm.ac.uk (V.Y.); 2Department of Pharmaceutics, UCL School of Pharmacy, University College London, London WC1N 1AX, UK; s.murdan@ucl.ac.uk; 3Department of Biology, York Biomedical Research Institute, University of York, York YO10 5DD, UK

**Keywords:** cutaneous leishmaniasis, *Leishmania major*, chitosan nanoparticles, amphotericin B, in vivo

## Abstract

Chitosan nanoparticles have gained attention as drug delivery systems (DDS) in the medical field as they are both biodegradable and biocompatible with reported antimicrobial and anti-leishmanial activities. We investigated the application of chitosan nanoparticles as a DDS for the treatment of cutaneous leishmaniasis (CL) by preparing two types of chitosan nanoparticles: positively charged with tripolyphosphate sodium (TPP) and negatively charged with dextran sulphate. Amphotericin B (AmB) was incorporated into these nanoparticles. Both types of AmB-loaded nanoparticles demonstrated in vitro activity against *Leishmania major* intracellular amastigotes, with similar activity to unencapsulated AmB, but with a significant lower toxicity to KB-cells and red blood cells. In murine models of CL caused by *L. major,* intravenous administration of AmB-loaded chitosan-TPP nanoparticles (Size = 69 ± 8 nm, Zeta potential = 25.5 ± 1 mV, 5 mg/kg/for 10 days on alternate days) showed a significantly higher efficacy than AmBisome^®^ (10 mg/kg/for 10 days on alternate days) in terms of reduction of lesion size and parasite load (measured by both bioluminescence and qPCR). Poor drug permeation into and through mouse skin, using Franz diffusion cells, showed that AmB-loaded chitosan nanoparticles are not appropriate candidates for topical treatment of CL.

## 1. Introduction

Cutaneous leishmaniasis (CL) is a neglected tropical disease (NTD), caused by intracellular parasites belonging to the genus *Leishmania*, which are transmitted among mammals through the bite of female sand flies [1,2]. CL affects around 1.5 million people worldwide every year [3]. CL lesions can heal spontaneously, in most cases within 2–18 months, and are not usually fatal but can cause considerable disfigurement, psychological disorders and social stigma leading to reduction in individual self-esteem [3,4]. All of the currently available drugs for CL-pentavalent antimonials, the aminoglycoside paromomycin, the alkylphospholipid miltefosine and the polyene antifungal amphotericin B (AmB)—have drawbacks such as, variable cure rates, toxicity and high costs. These limitations highlight the need for novel drugs or formulations that can provide a short, safe, efficacious, affordable and field-adapted treatment for all forms of cutaneous leishmaniasis [5]. The process of developing and discovering new drugs is long (more than 10 years), slow, expensive and challenging [5,6]. In addition, NTDs are generally considered commercially unattractive for pharmaceutical R & D programmes [7].

Drug delivery systems (DDS) can be used to increase efficacy and decrease toxicity of known drugs by modulating their pharmacokinetic properties and by enabling drug targeting to sites of infection [8,9]. Polymeric nanoparticles prepared by the ionotropic gelation method (formed by interactions between two oppositely charged entities), are of interest as DDS as they have advantages (over others such as liposomes, niosomes, emulsions) of lower cost, simple and quick preparation without the requirement for organic solvents, and have a long shelf life at room temperature [10]. Chitosan nanoparticles have garnered a lot of attention in the medical field as they are biocompatible and biodegradable and can be adapted for controlled release, all important properties for drug safety. They are increasingly being considered for a variety of biomedical applications, e.g., wound healing [11,12] and chitosan has been approved by The Food and Drug Administration (FDA) for use in wound dressings [13]. Chitosan nanoparticles of different sizes and charges, can be prepared by the ionotropic gelation method, and are suitable for different routes of administration [14].

For the potential treatment of infectious diseases, chitosan solution and nanoparticles have been shown to have activity against *Leishmania* parasites and other microbes (for example, *Staphylococcus aureus*, *Escherichia coli*, *Candida albicans)* [15,16,17,18,19]. Both chitosan solutions and nanoparticles have been reported to have the same antimicrobial mechanism, i.e., via interactions between the protonated NH_3_^+^ groups of chitosan and the negatively charged cell membrane of microbes [20]. We have previously demonstrated that chitosan solution (high molecular weight (HMW)) has a pH-dependent anti-leishmanial activity and this activity was not due to immune activation. Instead, the anti-leishmanial activity was related to direct chitosan pinocytic uptake into the parasitophorous vacuole (PV) in the host cell [21].

Intravenous amphotericin B deoxycholate (Fungizone™) is one of the few available second-line drugs for leishmaniasis. Amphotericin B forms pores in cell membranes, via complexation with ergosterol in *Leishmania* membranes [22]. However, the use of this deoxycholate amphotericin B is clinically limited due to infusion-related and nephrotoxicity issues [23,24]. A liposomal formulation (AmBisome^®^, diameter = 80 nm) [24,25] was developed to improve the tolerability and reduce the adverse events associated with conventional AmB administration. AmBisome^®^ is approved by the FDA for the treatment of visceral leishmaniasis (VL) and has also shown clinical efficacy in CL patients [24,26]. AmBisome^®^ is used widely for the treatment of VL as it is donated by Gilead via WHO but is only available at high cost for CL. In addition to the requirement for the cold chain [22,27,28], AmBisome^®^ has also limitations of a complex production process and an increase in particle size and a change in drug content upon storage have been reported during 72 h of storage at room temperature [28,29].

A number of different types of chitosan nanoparticles encapsulating AmB have been previously evaluated against *Leishmania* parasites with promising results in vitro and in vivo mainly in experimental models of VL, for example as nanoemulsion-based chitosan nanocapsules entrapping AmB (Asthana et al. 2013) against *L. donovani*-infected hamsters [30], and AmB-loaded chitosan-coated solid lipid nanoparticles (Jain et al., 2014) [31]. There have been far fewer studies on CL. Zadeh Mehrizi et al. (2019) reported a complete cure of lesions in *L. major* infected BALB/c mice by AmB-loaded chitosan nanoparticles prepared by a phase separation method [32]. These nanoemulsion-based nanocapsules, solid lipid nanoparticles and nanoparticles have drawbacks due to the use of organic solvents or heat during formulation preparation, the complexity of preparation and their low stability. The ionotropic gelation method for nanoparticles has been used once for studies on CL by using positively charged chitosan and negatively charged chondroitin sulphate to prepare AmB-loaded chitosan nanoparticles that were 136 ± 11 nm in diameter and positively charged with a zeta potential of +8.4 mV to +30.2 mV (Ribeiro et al. 2014) [15]. When these particles were tested in vivo in *L. amazonensis*-infected BALB/c mice, by intravenous administration at 1 mg of drug/kg daily for 10 days, the mice showed a smaller lesion on the footpad (around 25% reduction) which was sustained for up to 30 days following the end of the treatment, compared to mice receiving 1 mg of AmB for 10 days [33].

We aimed to improve the potential of chitosan nanoparticles for the treatment of CL through the first in-depth study on this disease using: (a) the ionotropic gelation method to prepare nanoparticles of both positive and negative charge, using sodium tripolyphosphate (TPP), an FDA-approved additive for use in food substances [34] which has shown to play a role in nanoparticle stability during storage [15,35] or dextran sulphate (biodegradable, biocompatible and previously used to produce chitosan nanoparticles [35]) for gelation, respectively, with a clear aim of preparing particles <100 nm with enhanced drug delivery to *Leishmania* infected skin [24], (b) to determine the activity and dose-response effect in vivo in a murine model of CL infection using three methodologies [36,37], (c) to relate activity to drug accumulation and distribution in the skin [22] and (d) to compare the activity of these formulations against CL when administered by both the intravenous and topical routes, with the topical formulations optimized and analysed for appropriate skin permeation and distribution [37].

## 2. Results

### 2.1. Particle Preparation and Physicochemical Properties

Freeze-drying of the particle suspension yielded a powder, which was white for blank particles and yellow for AmB-loaded nanoparticles. The yield, and drug encapsulation efficiencies (EE) were ≥90% for both AmB-loaded chitosan-TPP (AmB-CH-TPP) and AmB-loaded chitosan-dextran sulphate nanoparticles (AmB-CH-Dex) (Table 1). Drug loading was similar for both types of particles (around 25%, *p* > 0.05 by using *t*-test, Table 1).

Other properties of blank chitosan-TPP (CH-TPP), blank chitosan-dextran sulphate (CH-Dex), AmB-CH-TPP and AmB-CH-Dex nanoparticles are also shown in Table 1. Chitosan-TPP nanoparticles were smaller (approximately 2.5×) in size (≈70 nm) than chitosan-dextran nanoparticles (≈175 nm), although their polydispersity indices (PDI) were similar. Chitosan-TPP nanoparticles were positively charged (zeta potential ≈ +25 mV) while chitosan-dextran nanoparticles were negatively charged (zeta potential ≈ −12 mV). Drug loading caused no changes in particle size, polydispersity index or zeta potential (*p* > 0.05 by using *t*-test). Lyophilisation without a cryoprotectant resulted in poor quality nanoparticles with aggregation of the particles (PDI > 1). In contrast, the use of a cryoprotectant (sucrose 5%) during lyophilisation, led to good quality nanoparticles (PDI < 0.3) and increased particle size by 6–40% for both types of nanoparticles (Table 1, *p* < 0.05 by using *t*-test), while causing no change in the particle surface charge as shown by the zeta potential (*p* > 0.05 by using *t*-test).

Transmission (Appendix A) and scanning electron micrographs (SEM) of the lyophilised particles (Appendix A) showed all the particles (chitosan-TPP or chitosan-dextran sulphate nanoparticles) to be spherical. Drug loading caused no changes in particle shape. Stability of chitosan nanoparticles following incubation in different media is shown in Appendix A, both types of AmB-loaded chitosan nanoparticles did not show any significant change in their size or zeta potential at temperatures of 4, 34 and 37 °C when incubated in different media (water, PBS or RPMI at pH 7.5 or 6.5) or in mouse (BALB/c) plasma at 4 °C for a period of 30 days.

### 2.2. In Vitro Release of AmB from the Nanoparticles

AmB release profiles from chitosan-dextran and chitosan-TPP nanoparticles in PBS (pH 5, 6.5 or 7.5) and in mouse plasma at 37 °C are shown in Figure 1, indicating slow drug release from both types of nanoparticles. However, there was an influence of medium pH, with PBS pH 5 causing almost 2× greater release (38% and 51% of AmB was released from AmB-CH-Dex and AmB-CH-TPP nanoparticles, respectively after 196 h) than pH 6.5 or 7.5 (where 19% and 30% of AmB was released from AmB-CH-Dex and AmB-CH-TPP nanoparticles, respectively after 196 h at both pH 6.5 or 7.5) (Figure 1A,B, *p* < 0.05 by one-way-ANOVA). A difference in drug release was also seen for particle type, with greater drug release (1.5×) from chitosan-TPP compared to chitosan-dextran particles (Figure 1C, *p* < 0.05 by one-way-ANOVA). Similar findings were found at temperatures of 4 °C and 34 °C, data are shown in Appendix A.

### 2.3. In Vitro Activities of Chitosan Formulations against L. major and L. mexicana

The toxicity of AmB-loaded nanoparticles was significantly lower than that of the AmB solution (19× less toxic against RBC and 6× less toxic against KB cells, *p* < 0.05 by extra sum-of-squares F test). When the toxicities of these formulations were compared to that of AmBisome^®^, it was found that AmB-loaded chitosan nanoparticles had the same toxicity towards KB cells (*p* > 0.05 by extra sum-of-squares F test), but were more toxic to RBCs (2.5×) (See Appendix A for more details).

The EC_50_ and EC_90_ values of chitosan formulations and of controls against *L. major* and *L. mexicana* promastigotes and amastigotes are shown in Appendix A and Table 2, respectively. TPP, dextran and CH-Dex had no anti-leishmanial activity at concentrations up to 486 µg/mL, the maximum concentration used in the study. In contrast to CH-Dex nanoparticles, CH-TPP nanoparticles and the chitosan solution showed anti-leishmanial activity against both promastigotes and amastigotes with higher activity (6–15×) at the lower pH 6.5 than at pH 7.5. Both the chitosan solution and CH-TPP nanoparticles had a similar activity against intracellular *L. major* and *L. mexicana* amastigotes at a lower pH 6.5 (Appendix A and Table 2). Loading of AmB in both types of chitosan nanoparticles significantly increased anti-leishmanial activity against promastigotes and amastigotes by orders of magnitude (*p* < 0.05 by extra sum-of-squares F test), such that AmB-loaded chitosan nanoparticles had similar activities as AmB solution (EC_50_ ≈ 0.08 µg/mL and EC_50_ ≈ 0.3 µg/mL for both AmB solution and nanoparticles against *L. major* promastigotes and amastigotes, respectively, (*p* > 0.05 by extra sum-of-squares F test), and were more potent than AmBisome^®^ (EC_50_ ≈ 1 µg/mL and EC_50_ ≈ 1.2 µg/mL against *L. major* promastigotes and amastigotes, respectively *p* < 0.05 by an extra sum-of-squares F test; Appendix A and Table 2). When the two *Leishmania* species are compared, AmB-CH-TPP and AmB-CH-Dex nanoparticles, like the AmB solution, had 2–3× greater activity against *L. major* than *L. mexicana*, for both promastigotes and amastigotes (*p* < 0.05 by extra sum-of-squares F test) (Appendix A and Table 2).

### 2.4. Anti-Leishmanial Activity of Chitosan Formulations In Vivo

The effect of the formulations on: (i) lesion size on the mouse rump, (ii) parasite load measured by parasite-associated bioluminescence and (iii) parasite load quantified by qPCR are shown in Figure 2. AmB-CH-TPP nanoparticles (5 mg of AmB/kg/QAD for 10 days; i.v.) were the most effective formulation compared to the other chitosan and AmB formulations, causing reductions in lesion size and parasite load measured using bioluminescence and qPCR of 87%, 99% and 98%, respectively after the 10-day treatment. Such reductions were statistically similar to those caused by the positive control paromomycin (50 mg/kg/QD for 10 days; i.p., *p* > 0.05 by one-way-ANOVA-post Tukey) but significantly greater (around 2×) than those caused by AmBisome^®^ (10 mg of AmB/kg/QAD for 10 days; i.v., *p* < 0.05 by one-way-ANOVA-post Tukey) (Figure 2). Blank CH-TPP nanoparticles were as effective as AmBisome^®^ at reducing lesion size (by 34% vs. 36%), parasite load (parasite-associated bioluminescence; by 62% vs. 72%) and parasite load (qPCR; by 67% vs. 68%) (*p* > 0.05 by using one-way-ANOVA) (Figure 2). In contrast to CH-TPP nanoparticles, CH-Dex nanoparticles caused no changes in lesion size or parasite load measured by bioluminescence or qPCR compared to the untreated controls (*p* > 0.05 by using one-way-ANOVA, Figure 2). The influence of loading AmB into chitosan-dextran nanoparticles could not be determined as 24 h following after the first (and only) dose of the formulation, the mice looked unwell with piloerection and weight loss. Two mice had died after two days, without any signs of CL-related mortality such as severe ulceration and dissemination of the lesion.

In mice dosed with AmB-containing formulations, drug concentrations in the infected lesion (rump skin) and control skin (uninfected skin on back) were determined at the end of the experiment (at day 10) and are shown in Appendix A. AmB concentrations in the lesion were higher than in the control non-lesion skin (between 13 and 20-fold) for both mice groups treated with AmB-CH-TPP nanoparticles and AmBisome^®^. AmB-CH-TPP nanoparticles (5 mg/kg/QAD for 10 days; i.v.) resulted in 44.6 μg/g intralesional AmB which was 6.8× greater than drug levels achieved by AmBisome^®^ (10 mg/kg/QAD for 10 days; i.v.) (Figure 3).

### 2.5. The Dose Dependency of Anti-Leishmanial Activity of AmB-Loaded Chitosan-TPP Nanoparticles in L. major-Infected Mice

AmBisome^®^ and AmB-CH-TPP nanoparticles at AmB doses of 1.25, 2.5 and 5 mg per mouse weight (kg) were administered intravenously on alternate days for 10 days. The reductions of lesion size, and parasite load measured using bioluminescence and qPCR were measured at the end of the experiment (at day 10). CH-TPP nanoparticles and paromomycin treatment groups were also included. AmB-CH-TPP nanoparticles (5 mg/kg AmB dose) was as effective as paromomycin (*p* > 0.05 by using one-way-ANOVA) in causing reductions of lesion size (83% vs. 89%, respectively), parasite load measured by bioluminescence signal (99% for both) and by qPCR (98% vs. 99%, respectively) compared to the untreated controls (Figure 4).

Lower concentrations of AmB (i.e., 2.5 mg/kg and 1.25 mg/kg) in chitosan-TPP nanoparticles resulted in lower efficacies compared to the 5 mg/kg dose in the same formulation ((*p* < 0.05 by one-way ANOVA)). However, these lower doses of AmB chitosan-TPP nanoparticles achieved statistically similar efficacies compared to AmBisome^®^, as shown in Table 3 and Figure 4 (*p* > 0.05 by one-way ANOVA).

The relationship between AmB dose and the different formulations on drug concentrations in the infected lesion (rump skin) and in the control skin (uninfected skin on mouse back) are shown in Figure 5 and Table 4. After multiple doses of the formulations (AmBisome^®^ or AmB-CH-TPP nanoparticles), intralesional AmB levels were about 7-fold lower in the AmBisome^®^ group compared to the group which received AmB-CH-TPP nanoparticles at 5 mg/kg of AmB (*p* < 0.05 by one-way-ANOVA), were similar to the group which received 2.5 mg/kg of AmB in chitosan-TPP nanoparticles (*p* > 0.05 by one-way-ANOVA) and were higher than the group 6 which received 1.25 mg/kg of AmB in chitosan-TPP nanoparticles (*p* < 0.05 by one-way-ANOVA). For all groups dosed with AmB formulations, drug levels in lesions were significantly higher (around 20×) than in the control uninfected skin sites (Table 4; *p* < 0.05 by one-way-ANOVA). No AmB was detected in skin samples from the untreated group and the positive control (paromomycin group).

The intralesional AmB levels related to the drug doses used and to the response (parasite load measured by bioluminescence and qPCR) are shown in Figure 5, including a nonlinear-fit sigmoidal dose–response curve with logarithm of intralesional AmB levels versus reductions in lesion sizes and parasite load, compared to the untreated controls.

Correlations between drug dose and relative reductions in lesion sizes, and parasite load measured using bioluminescence and qPCR were high and statistically significant (r values = 0.9 of Pearson correlation coefficients and *p* < 0.05 by using one-way-ANOVA). Effective dose (ED_50_ and ED_90_ values) were calculated after plotting the log dose used against percentage response and are shown in Table 5.

### 2.6. Ex Vivo Permeability of Leishmania-Infected Skin to AmB in Different Formulations

The permeability of uninfected and *L. major*-infected skin to AmB-loaded chitosan nanoparticles were evaluated using Franz diffusion cells. The cumulative concentration of AmB from AmB-CH-TPP and AmB-CH-Dex nanoparticles in the receptor compartment of Franz diffusion cells as a function of time is shown in Figure 6. When applied as a solution, AmB did not permeate through either uninfected or infected skin throughout the 24 h experiment. In contrast, when applied as AmB-CH-TPP nanoparticles (Size = 65 ± 8 nm, Zeta potential = 25.5 ± 1 mV) or AmB-CH-Dex nanoparticles (Size = 170 ± 8 nm, Zeta potential = −13 ± 1 mV), AmB could be detected in the receptor phase. At the end of the 24 h experiment, both types of AmB-loaded chitosan nanoparticles showed approximately a two-fold higher permeation of AmB through *Leishmania*-infected skin compared to uninfected control skin (*p* < 0.05 by repeated measures ANOVA). Furthermore, approximately twice as much AmB permeated into the receptor phase from chitosan-TPP nanoparticles compared to from chitosan-dextran nanoparticles, through both uninfected and infected skin (*p* < 0.05 by repeated measures ANOVA).

Lag time, flux and permeability coefficients of the formulations are shown in Table 6. There was no significant difference in the lag time between uninfected and infected skin for both types of AmB nanoparticles (*p* > 0.05 by *t*-test) or between the two types of chitosan nanoparticles (*p* > 0.05 by *t*-test) (Table 6). The flux was 2× higher for both types of AmB-loaded chitosan nanoparticles in infected skin compared to uninfected skin. The permeability coefficient was 1.8 and 2.5× higher for AmB-CH-TPP nanoparticles and AmB-CH-Dex nanoparticles, respectively in infected skin compared with uninfected skin (Table 6). All the above indicated that *L. major* infection of the skin enhanced the permeation of both types of nanoparticles. However, although permeation by nanoparticles is detected, it is both slow and limited.

At the end of the permeation experiment, drug concentration in the skin samples was measured (Table 7). When AmB solution was used, more than 90% of the drug remained on the skin surface while 5–8% was extracted from the skin itself and no drug was detected in the receptor phase. Similarly, for the AmB-loaded chitosan nanoparticles, the majority of the drug (61% of drug from AmB-CH-TPP nanoparticles and 73% of drug from AmB-CH-Dex nanoparticles) remained on the infected skin surface, followed by drug within this skin (38% of AmB-CH-TPP nanoparticles and 26% of AmB-CH-Dex nanoparticles), and only a limited amount (0.1–0.4% of applied drug) passed through the skin to enter the receptor phase, with greater permeation through infected skin than uninfected skin (2×) and from AmB-CH-TPP nanoparticles compared to AmB-CH-Dex nanoparticles (4×).

In separate experiments, skin sections were examined by fluorescence microscopy following the 24 h permeation experiment using uninfected and *L. major*-infected mouse skin. Micrographs (Appendix A) showed no penetration of rhodamine-labelled chitosan-TPP nanoparticles (Size = 72 ± 7 nm, Zeta potential = 22 ± 2) or of rhodamine-labelled chitosan-dextran sulphate nanoparticles (Size = 174 ± 7 nm, Zeta potential= −14 ± 2) or of rhodamine labelled chitosan solution.

## 3. Discussion

In our previous study (Riezk et al. 2020), we described how chitosan (HMW) was active in vitro against intracellular *L. major* amastigotes [21]. To translate this activity into an in vivo model of *L. major* infection, we used chitosan nanoparticles prepared by the ionotropic gelation method as a drug delivery vehicle for AmB because chitosan nanoparticles: (i) potentially reduce the toxicity of AmB, improve its efficacy, modulate AmB pharmacokinetics, permit sustainable AmB release at the site of infection and protect the drug from degradation [15,33,38], (ii) have promising features for DDS due to their biocompatibility, biodegradability, controlled drug release, mucoadhesiveness, wound healing and antimicrobial properties [11,12,39] and (iii) have high stability at different temperatures and a simple preparation process [40,41] while other drug carriers, such as liposomal formulations, need the cold chain for stability, have a complex preparation process and can be costly which limits their use [22,27,29]. For the synthesis of chitosan nanoparticles, we selected two crosslinkers: TPP which resulted in positively charged nanoparticles and dextran sulphate which resulted in negatively charged nanoparticles.

To maximise the possibility of anti-leishmanial activity of our chitosan particles, we aimed to synthesise the smallest size nanoparticles possible, as nanoparticles of diameter under 100 nm have shown a 2.5-fold higher uptake in Caco-2 cells compared to 1 µm sized particles and a 6-fold higher uptake than 10 µm sized particles [42]. Of greater relevance to this study, AmB liposomes with size < 100 nm showed extra-vasation at the inflammatory site of infection [22]. Additionally, small-sized nanoparticles have exhibited maximum deposition of their content in the skin dermis following topical application [43], and intravenously administered small nanoparticles have facilitated targeting of macrophages residing in the skin [43,44,45,46]. Small-sized nanoparticles (<100 nm) have also been reported to enable greater cell uptake, skin permeation and immune-targeting—all desirable properties in formulations for topical CL treatment [47,48]. Of the chitosan nanoparticles prepared for our study, those with dextran were significantly larger than the ones prepared with TPP, as also reported by Raja et al. (2015). This could be explained as TPP is a small polyanionic molecule that forms strong ionic interactions with the NH^3+^ groups of chitosan [49]. The encapsulation efficiency of AmB in both types of nanoparticles was similar to other reports of AmB encapsulation in chitosan nanoparticles with TPP (80%) or with chondroitin sulphate (90%) [15,40].

Both chitosan-TPP and chitosan-dextran nanoparticles released the encapsulated drug slowly with greater drug release at the lower pH of 5 than at pH of 7.5, probably due to the higher solubility of chitosan in acidic media [50]. This indicates that the entrapped drug is released by particle dissolution in the suspending medium. Chitosan-TPP nanoparticles released the drug faster than chitosan-dextran nanoparticles. This could be explained by the particle size; as AmB-CH-TPP nanoparticles were 2.5× smaller than AmB-CH-Dex nanoparticles, the surface area to volume ratio of chitosan-TPP nanoparticles is 3× larger allowing greater AmB release from the particle surface [42]. In addition, the stability of the size and charge of nanoparticles and slow release of encapsulated drug in plasma would minimise the likelihood of initial burst-release and related AmB toxicity. Their observed stability in various media supported further in vivo studies of chitosan-TPP and chitosan-dextran sulphate nanoparticles as DDS for AmB.

In our initial in vitro studies, we showed that AmB-loaded chitosan nanoparticles showed similar anti-leishmanial activity to the AmB solution and were less toxic against RBC and KB-cells. Loading AmB into chitosan nanoparticles mitigated its serious toxicity against RBC and KB-cells, presumably by entrapping and retaining the AmB, while allowing slow drug release. Similar findings have been reported for AmB-loaded chitosan-chondroitin sulphate nanoparticles, which showed less toxicity against RBC and murine macrophages than AmB solution [15]. Lowering the pH of RPMI medium from 7.5 to 6.5 increased the anti-leishmanial activity of chitosan solution and CH-TPP nanoparticles against *L. major* and *L. mexicana* promastigotes and amastigotes by 7–20× due to the greater chitosan ionisation at lower pH for both chitosan solution and CH-TPP nanoparticles (positive surface charge) [21]. AmB-loaded chitosan nanoparticles showed a similar activity against *L. major* and *L. mexicana* promastigotes and amastigotes to AmB solution and significantly higher activity than AmBisome^®^ at the two pH values. Ribeiro et al. (2014) reported that the anti-leishmanial activity of AmB-loaded chitosan-chondroitin sulphate nanoparticles (136 ± 11 nm, positive charge) was similar to AmB solution in studies on *L. amazonensis* and *L. chagasi* promastigotes with similar EC_50_ values to our study [33]. Additionally, our EC_50_ values against *L. major* and *L. mexicana* amastigotes were in accordance with another report that found the EC_50_ values of chitosan-coated AmB-loaded solid lipid nanoparticles (158.9 ± 7.1 nm, positive charge), AmBisome^®^ and deoxycholate AmB were 0.022 ± 0.07, 0.086 ± 0.04 and 0.253 ± 0.03 μg/mL, respectively, against *L. donovani* amastigotes infecting the mouse macrophage cell line J774 A.1 after 72 h of incubation [31]. Ribeiro et al. (2014) showed that a chitosan solution (M_W_ = 60–120 KDa) had EC_50_ values of 66 ± 1 and 71 ± 1 µg/mL and blank chitosan nanoparticles had EC_50_ values of 52 ± 2 and 46 ± 6 µg/mL against *L. amazonensis* and *L. chagasi* promastigotes, respectively. These values are different from EC_50_ values in our study (Table 2 and Appendix A). This difference could be explained by the use of (i) different *Leishmania* species, (ii) 48 h incubation of compounds with *Leishmania* and (iii) pH of the medium (could have been different but they did not mention theirs) [33].

The main focus of our studies was to determine the efficacy of the chitosan formulations against *L. major* with dose response and three parameters to measure the pharmacodynamic (PD) properties of these particles. In our previous studies [22], we have also related efficacy to pharmacokinetic (PK) parameters in particular accumulation in normal skin and the lesion (on the rump). In addition, we intended to compare administration both (i) intravenously, which is the fastest route as AmB nanoparticles are delivered immediately into the blood stream enabling a high bioavailability [51], and (ii) topically because of the ease of administration and reduced systemic side effects of topical formulations over systemic therapy [52]. We evaluated both routes, topical and systemic treatment as topical treatment can be used in cases with localized CLwhile systemic treatment is recommended in patients with complex CL [52].

CL skin infection plays an important role in drug accumulation, as shown for AmB accumulation in the localized lesion [22], with the localized inflammatory immune response caused by *L. major* parasites replicating within dermal macrophages of CL having an important role. At the site of infection, the leaky vasculature could enhance the permeation and retention of the drug which loads to local drug accumulation [22,53]. Another explanation could be that phagocytic monocytes migrate from the bloodstream to the infection site (skin lesion) acting as Trojan Horse for drug delivery [54,55,56]. Similar findings were reported by Wijnant et al. (2018) as AmB levels were 5- to 20-fold higher in *L. major* infected BALB/c mouse skin than in healthy skin from same infected mice following dosing with AmBisome^®^ or AmB deoxycholate [22]. AmB-CH-TPP nanoparticles (5 mg/mL/QAD for 10 days; i.v.) resulted in significantly higher levels of AmB accumulation in infected skin than AmBisome^®^ (10 mg/kg/QAD for 10 days; i.v.), reflecting reports by Shahnaz et al. (2017) who showed that mannose-anchored thiolated chitosan amphotericin B nanoparticles (Size = 362 nm, Zeta potential = 20 mV) showed 73 and 23 fold increase in uptake by *L. donovani* infected J774 macrophages compared with AmB solution and AmBisome^®^, respectively [57].

Blank CH-TPP particles showed a similar efficacy to AmBisome^®^ in our in vivo studies. This is similar to findings of Ribeiro et al. (2014) who reported that intravenously administered blank chitosan-chondroitin sulphate nanoparticles (104 ± 11 nm, positive charge) caused a significant reduction in lesion size on footpad of *L. amazonensis* infected BALB/c mice [33].

In our studies, AmB-CH-TPP nanoparticles (5 mg/mL/QAD for 10 days; i.v.) showed a similar efficacy to the positive control (paromomycin, 50 mg/kg/QD for 10 consecutive days; i.p). The superior efficacy of AmB-CH-TPP nanoparticles (5 mg/mL/QAD for 10 days; i.v.) compared to AmBisome^®^ (10 mg/kg/QAD for 10 days; i.v.) could be related to both the higher intralesional AmB concentrations (described above, Table 3) and the activity of chitosan nanoparticles themselves against CL. Ribeiro et al. (2014) reported that AmB-loaded chitosan-chondroitin sulphate nanoparticles (136 ± 11 nm, positive charge) caused significant reductions (around 25%) in the lesion size and in the parasite burden of *L. amazonensis* infected BALB/c mice, when administrated intravenously (1 mg/kg/day for 10 days), compared to mice receiving 1 mg of AmB for 10 days [33]. The significant correlation between intralesional AmB level and therapeutic outcome of AmB-CH-TPP nanoparticles, along with the anti-leishmanial AmB concentration-dependent response is consistent with previous reports using AmBisome^®^ in the same model of infection [22].

Topical treatment offers several advantages over systemic treatment in terms of side effects, direct targeting to infected (skin) lesions, a reduced need for patient follow up and potentially improved patient compliance [58,59]. As mentioned above, our initial aims included the development of topical nanoparticles formulations (positive and negative charged nanoparticles) of AmB. This was based upon four fundamental considerations that contribute to the efficacy of topical treatment of CL:(i)The intrinsic efficacy of the (AmB solution and AmB-loaded chitosan nanoparticles) against Old World (*L. major*) and New World cutaneous *Leishmania (L. mexicana)*,(ii)The permeability of the (AmB solution and AmB-loaded chitosan nanoparticles) through the skin to reach the dermis where the *Leishmania* infected macrophages reside [52],(iii)Disposition of the drug in the skin [60],(iv)The release of the active compound from the formulation into the infected macrophages or into the dermis of infected skin [59].

Both AmB-CH-TPP nanoparticles and AmB-CH-Dex nanoparticles showed high in vitro activities against *Leishmania* amastigotes. Therefore, as a first step we investigated their ex vivo permeation through uninfected and *L. major* infected mouse skin using Franz diffusion cells. As previously reported [61,62,63,64], AmB in aqueous solution did not permeate through uninfected or *L. major* infected skin, most likely due to its size (924 Da) and insolubility in water [61]. This impermeability of AmB through uninfected and infected skin explained the unsuccessful treatment in a different study after the topical application of AmB in on *L. major* infected mice [65]. Our ex vivo permeation study showed a limited and slow permeation of AmB through healthy and infected mice skin when both types of AmB-loaded chitosan nanoparticles were applied on the skin, with a long lag time of about 20 h indicating a long time for the steady state flux to be reached (which reflects slow permeation across the stratum corneum). These data were confirmed by imaging the permeation of rhodamine labelled chitosan-TPP nanoparticles and rhodamine labelled chitosan-dextran sulphate nanoparticles across uninfected and *L. major* infected skin by fluorescence microscopy which showed that these nanoparticles stayed on the skin. Our results were consistent with other reports; Vogt et al. reported that most of applied 42–300 nm fluorescent silica nanoparticles stayed in the upper layers of the excised human skin using conventional fluorescence microscopy of skin sections [66].

Similar observations regarding the limited permeation of chitosan nanoparticles were reported by Nair et al., when curcumin-encapsulated chitosan nanoparticles with sizes ranging from 167.3 ± 3.8 nm to 251.5 ± 5.8 nm permeated very slowly through Strat-M^®^ membrane (Strat-M is made of polyester sulfone arranged as multiple layers mimicking the skin structure including a tough outer layer manufactured by Merck) using Franz diffusion cells [67]. Malli et al. (2019) reported that the daily topical application of chitosan-coated Poly (isobutyl cyanoacrylate) nanoparticles (Size = 187 nm, Zeta potential = 53.8 mV, prepared by anionic emulsion polymerization method) gelled by Pluronic F127 for 3 consecutive weeks to BALB/c mice infected with *L. major*, resulted in partial and incomplete healing of lesions; the nanoparticles were not effective enough to heal the lesion [68].

Our Franz diffusion study showed that AmB-loaded chitosan nanoparticles offered greater permeation of AmB through infected than through uninfected skin, consistent with a previous publication showing greater permeation of caffeine and ibuprofen through *L. major* infected compared to uninfected BALB/c mouse skin [61]. This same study reported no permeation of paromomycin sulphate through uninfected mouse skin but a high permeation through *L. major* infected skin [61]. CL pathology causes damage to the skin barrier and this alteration could enhance the penetration of nanoparticles [61]. *Leishmania* infected skin is characterised by the presence of abundant inflammatory cells in the infection site and this could also disrupt the architecture of the epidermal and dermal skin layers such as ulceration and necrosis [61,64,69]. Van Bocxlaer et al. (2016) reported that trans-epidermal water loss (TEWL) was significantly higher in *L. major* infected skin and this reduced the barrier function of the skin and subsequently increased the accumulation of fluid in the interstitial spaces causing oedema that could enhance the permeation of water-soluble compounds [61].

In our study, AmB-CH-TPP nanoparticles (Size = 68 ± 7 nm, Zeta potential = 30 ± 2) presented greater permeation of AmB than AmB-CH-Dex nanoparticles (Size = 168 ± 7 nm, Zeta potential = −15.5 ± 2). Similarly, Try et al. (2016) reported that smaller poly(L-lactide-co-glycolide) nanoparticles (70 nm) penetrated healthy male Swiss mice skin further than larger particles (300 nm) [70]. Another explanation of the greater penetration is that the positive surface charge of AmB-CH-TPP nanoparticles could interact with negative charges in the skin promoting close interaction and an occlusive barrier that could cause hydration, which would facilitate nanoparticle permeation through the skin [71,72]. The limited and slow skin permeation of AmB from AmB-loaded nanoparticles suggests these nanoparticles are acting as DDS and release the AmB, which then permeates into and through the skin (rather than carrying the encapsulated AmB into the skin). The limited skin permeation of AmB shows that the nanoparticles might be unsuitable candidates for topical administration and in vivo evaluation was not pursued further.

In summary, AmB-CH-TPP nanoparticles showed anti-leishmanial dose–response activity, stability and target-oriented drug delivery in an experimental model of cutaneous leishmaniasis. When administered by the i.v. route, these nanoparticles were significantly more active than AmBisome^®^ in the murine model (female BALB/c mice) of *L. major*, even when the drug doses were lower in the nanoparticles, with activity related to skin accumulation. AmB-CH-TPP nanoparticles can specifically target the CL lesions to a greater extent than AmBisome^®^ resulting in a higher concentration of AmB in lesion sites. In contrast to the successful parenteral route, the topical route of application was not pursued due to poor drug permeation into and through the mouse skin, which suggested that chitosan nanoparticles are not an appropriate candidate for topical treatment for CL. Our results indicate the need for further studies with amphotericin loaded chitosan-TPP nanoparticles, using the promising intravenous route and a range of *Leishmania* species, mammalian models and extensive toxicity studies.

## 4. Materials and Methods

### 4.1. Drugs and Chemicals

Amphotericin B (Purity ≥ 95%, Cambridge Bioscience, Cambridge, UK) was dissolved in DMSO at 10 mM as a stock solution and diluted to final concentrations in RPMI-1640 (Sigma, Gillingham, UK) with 10% (*v*/*v*) heat-inactivated foetal calf serum (HiFCS, Gibco, UK), referred as AmB solution.

AmBisome^®^ (a liposomal formulation of AmB, Gilead Sciences International Ltd., UK) was prepared according to the manufacturer’s instructions. Briefly, a suspension of AmB liposome was prepared in cold sterile MilliQ water to obtain an initial concentration of 4 mg/mL of AmB. The suspension was shaken and incubated at 65 °C for 10 min and then cooled to room temperature. Further dilution to the required concentration of AmBisome^®^ was done with 5% dextrose (aq) (*w*/*v*) [22].

A solution of HMW chitosan (M_W_ = 310–375 KDa, Sigma) was prepared by dissolving 1 g in 100 mL of 1% (*v*/*v*) acetic acid solution (Sigma) at room temperature with continuous stirring for 24 h until a clear solution was obtained. The pH of the solution was adjusted to approximately 6.5 by adding sodium hydroxide solution (2 M, NaOH, Sigma) using a pH meter (Orion Model 420 A, Thermo Fisher Scientific, Waltham, MA, USA).

### 4.2. Mice and Ethics Statement

Female BALB/c mice aged 6 to 8 weeks, at 18–20 g, were purchased from Charles River Ltd. (Margate, UK). These mice were kept in controlled rooms with humidity of 55% and temperature of 26 °C and fed water and rodent food ad libitum.

All animal work was carried out under a UK Home Office project licence according to the Animal (Scientific Procedures) Act 1986 and the new European Directive 2010/63/EU. The project licence (70/8427) was reviewed by the LSHTM Animal Welfare and Ethical Review Board prior to submission and consequent approval by the UK Home Office.

### 4.3. Parasites

Two *Leishmania* species: *L. major* (MHOM/SA/85/JISH118) and *L. mexicana* (MNYC/BZ/62/M379) were used. Promastigotes were cultured in Schneider’s insect medium (Sigma) with 10% (*v*/*v*) HiFCS, containing 200 units penicillin and 0.2 mg streptomycin per mL (PenStrep; Sigma). Promastigotes were incubated at 26 °C; the maximum in vitro passage number used was 5.

### 4.4. Preparation of Blank and AmB-Loaded Chitosan Nanoparticles

Then, 10 mL of TPP or dextran sulphate aqueous solution (without AmB for blank nanoparticles, or with 0.5 mL of AmB solution for drug-loaded particles) was added dropwise into the chitosan solution (10 mL) under magnetic stirring over a period of 5 min. Immediately afterwards, the nanoparticles suspension was sonicated for 15 min using a probe sonicator (Soniprep 150, Richmond Scientific Ltd., Chorley, UK, with the instrument set at 50% of its maximum output) to reduce the particle size. Subsequently, the nanoparticle suspension was filtered through a 0.2 µm syringe filter (Sigma, UK) to remove aggregates and larger particles. The nanoparticles were purified and concentrated by centrifugation (8000× *g*) using high recovery centrifugal filters (Spin-X UF concentrators, 30 kDa, Corning, Sigma, UK). This removed any unentrapped AmB from the particle suspension as only the small AmB molecule (M_W_ < 30 kDa) could pass through the membrane.

The nanoparticle suspension was then lyophilised in a freeze dryer (Micro Modulyo, Richmond Scientific Ltd.) using sucrose 5% *v*/*v* as a cryoprotectant to protect the nanoparticles from the freezing and desiccation stresses [73]. After 48 h, lyophilized nanoparticles were collected, weighed and stored at 4 °C for further analysis.

All nanoparticles—blank chitosan-TPP (CH-TPP), blank chitosan-dextran sulphate (CH-Dex), AmB-loaded chitosan-TPP (AmB-CH-TPP) and AmB-loaded chitosan-dextran sulphate (AmB-CH-Dex)-were prepared using the optimized parameters; 10 mL of HMW chitosan solution (30 mg in 10 mL of acetic acid (AC) 1%) at pH 5, 10 mg of AmB dissolved in 0.25 mL of DMSO (AmB solution), sonication time of 15 min and 10 mL of TPP solution (M_W_ 367.85 g/mol, Fisher scientific, US, 6 mg in 10 mL of double distilled water) or 10 mL of dextran sulphate solution (M_W_ 40 kDa, Sigma, 30 mg in 10 mL of double distilled water).

### 4.5. Determination of Drug Encapsulation Efficiency and AmB Loading

Following sonication, filtration and centrifugation (* as described in Section 4.4 above) above of prepared chitosan nanoparticles, the filtrates and supernatants were collected and analysed for AmB concentrations by HPLC as described below in Section 4.8. The encapsulation efficiency (EE), drug loading per particle mass [15] and nanoparticle yield [74] were calculated using the following equations:(1)$$EE\left(\%\right)=100\;\times \;\frac{\left(weight\;of\;total\;AmB-weight\;of\;free\;AmB\right)}{weight\;of\;total\;AmB}$$
(2)$$Yield\;\left(\%\right)=100\;\times \;\frac{weight\;of\;nanoparticles\;after\;freeze\;drying}{\left(weight\;of\;chitosan\;+\;weight\;of\;dextran\;or\;TPP\right)}$$
(3)$$Drug\;Loading\;\left(\%\right)=100\;\times \;\frac{\left(weight\;of\;total\;AmB-weight\;of\;free\;Amb\right)}{\left(weight\;of\;chitosan+weight\;of\;dextran\;or\;TPP\right)}$$

### 4.6. Physicochemical Properties (Size, Charge and Morphology) and Stability of the Nanoparticles

The physical characteristics of nanoparticles, including the average diameter (D), zeta potential (ZP), and polydispersity index (PDI), were measured using a ZetaSizer (Malvern Instruments Ltd., Malvern, UK) [43,75]. Data analysis was performed using the Malvern ZetaSizer software v 7.11. The morphology of the nanoparticles was examined by scanning electron microscopy (SEM) and transmission electron microscopy (TEM) at UCL, School of Pharmacy (see Appendix A for more information). For the SEM, a sample of nanoparticles was placed onto a self-adhesive carbon disc mounted on a 25 mm aluminium stub. The stub was coated with 25 nm of gold using a sputter coater and placed into a FEI Quanta 200 FEG SEM for imaging at 5 kV accelerating voltage using secondary electron detection. Nanoparticles were prepared as described in Section 4.4 and to measure their stability, nanoparticles were placed in 2 mL of distilled water, phosphate buffered saline (PBS) 0.9% NaOH (pH 7.4); Sigma) or RPMI (at pH 7.5 or pH 5) or in mouse (BALB/c) plasma (pooled female, BioIVT, West Sussex, UK) in rubber-capped glass vials at temperatures of 4, 34 or 37 °C for 30 days, and measuring particle size and zeta-potential after 1, 7 and 30 days of incubation.

### 4.7. In Vitro Release of AmB from Nanoparticles

The AmB release from the nanoparticle formulations was evaluated for up to 7 days. Although this does not mimic the in vivo situation, for which a comprehensive investigation would be necessary, it is in line with the longer dosing periods used. The standard in vivo drug efficacy for CL has been established as dosing over a 10-day period—where daily systemic administration with paromomycin demonstrated a significant reduction in nodule size and parasite load [36,37]. The release of AmB from AmB-CH-TPP nanoparticles or AmB-CH-Dex nanoparticles was conducted using dialysis. Then, 1 mL of the nanoparticles suspension (equivalent to 1 mg of AmB) in double distilled water was added to either 1 mL of PBS containing 5% DMSO (latter to ensure AmB solubility and sink conditions) [74] or 1 mL of mouse (BALB/c) plasma (pooled female, BioIVT) containing 5% DMSO. Subsequently, the 2 mL mixture was placed in a dialysis bag (molecular mass cut off = 12−14 kDa, Sigma) and dialyzed against 50 mL of PBS containing 5% DMSO at pH of 7.5, 6.5 or 5. After immersing the dialysis bag in the release medium, the dialysis set up was left to stir at 4, 34 or 37 °C for 7 days (168 h). The temperatures 4, 34 and 37 °C were chosen to mimic the storage, skin and body temperatures, respectively, while pH 5 was chosen to simulate the environment in the endosomal compartment of macrophages, pH 7.4 to simulate physiological conditions [76] and pH 6.5 to mimic our in vitro study (anti-leishmanial activity) conditions.

After 6, 24, 48, 72, 96, 120, 144 and 168 h, the total dialysis medium was replaced with fresh medium to avoid saturation of AmB (maintaining strict sink conditions throughout the experiment). Release media was processed to quantify the released AmB using HPLC as described below in Section 4.8. The results were expressed as a cumulative percentage release of the total amount of AmB (%*w*/*w*) versus time according to the equation.
(4)$$Cumulative\;release\;\left(\%\right)=100\;\times \;\frac{Weight\;of\;released\;AmB\;at\;time\;t}{Weight\;of\;total\;AmB}$$

### 4.8. Quantification of AmB by HPLC

AmB was analysed using a 1260 Infinity Agilent HPLC system. The column and settings used in our study are summarized in Table 8 as previously reported in [74]. A stock solution of AmB was prepared by dissolving 1 mg of AmB in DMSO. Standard solutions were prepared by diluting this stock solution in PBS containing 5% DMSO.

### 4.9. In Vitro Cytotoxicity Assay against Red Blood Cells

Human blood samples (EDTA anticoagulant, blood group O^+^, BioIVT) were centrifuged at 500× *g* for 5 min and the plasma aspirated and discarded. The remaining red blood cells (RBCs) were then washed three times in buffered saline (10 mM Tris, 150 mM NaCl, pH 7.4). The RBCs were then suspended in PBS to a density of 5 × 10^8^ cells/mL and exposed to 1000, 500, 250, 125, 65.5, 31.25, 15.62 and 7.81 μg/mL of chitosan formulations (chitosan solution, blank and AmB-loaded nanoparticles (AmB equivalent)) and AmB solution in 96 well plates (200 μL in each well) for 1 h at 37 °C. Lysis of RBCs was determined spectrophotometrically at 540 nm. Then, 20% Triton X-100 in PBS (Sigma) and phosphate buffer were used as the positive (representing 100% haemolysis) and negative controls, respectively. The results were expressed as the percentage reduction in human red blood cells compared with non-treated control wells, and represented by the 50% haemolytic concentration (RBC_50_) [77].

### 4.10. In Vitro Cytotoxicity Assay against KB Cells

Cytotoxicity of chitosan formulations against KB cells (human squamous carcinoma adherent cells derived from epidermal carcinoma from the mouth; ATCC [21]) was measured using a cell viability assay and resazurin sodium salt solution in 96 well plates (Sigma) as previously described by Riezk et al. (2020) [21]. Cytotoxicity was measured in RPMI 1640 at two pH values (at normal pH of RPMI 7.5 and at a lower pH 6.5). Podophyllotoxin (Sigma) was used as a positive control at a starting concentration of 0.05 μM.

### 4.11. In Vitro 72-h Activity of Chitosan and Its Derivatives against L. major and L. mexicana Extracellular Promastigotes and Intracellular Amastigotes

The activity of chitosan formulations against *L. major* and *L. mexicana* was evaluated in RPMI 1640 at two pH values (7.5 and 6.5). Anti-promastigote activity was evaluated by a cell viability assay using the resazurin sodium salt solution (Sigma) as previously described [21]. Results were expressed as percentage inhibition = 100% − x% viability (means ± SD) [21].

Peritoneal mouse macrophages (PEMs) were used as the macrophage host for intracellular amastigotes. The anti-leishmanial activity of formulations was expressed as the percentage reduction in the number of infected macrophages compared to untreated control wells, as previously described [21]. pH plays a critical role in the solubility and protonation of chitosan, so its activity against promastigotes and amastigotes was evaluated at two different pH values (pH = 7.5 and a lower pH of 6.5 by the addition of 0.05 M acidic buffer, 2-*N*-morpholino ethanesulfonic acid (MES, Sigma). RPMI 1640 plus MES (0.05 M) with pH = 6.5 had no activity against *Leishmania* [21].

### 4.12. Evaluation of the In Vivo Anti-Leishmanial Activity of Chitosan Formulations

Firstly, a dose-finding study was conducted in mice to determine the safe dose that could be used for the different formulations. This was found to be equivalent of 5 mg/kg to AmB when delivered by AmB-loaded chitosan–TPP particles and 10 mg/kg to AmB for AmB-loaded chitosan-dextran sulphate nanoparticles.

#### 4.12.1. In Vivo *L. major* Model of CL

Mice were shaved on the rump above the tail and subcutaneously infected with 200 μL of 4 × 10^7^ of stationary-phase luciferase-expressing *L. major JISH118 (Ppy RE9 H+L. major JISH118)* promastigotes in Schneider’s insect medium, this strain was kindly provided by Professor Louis Maes (University of Antwerp, Belgium). After 7 days, small nodules started to appear at the site of injection and the lesion size was recorded daily as the average of the width and length of the lesion using a digital calliper. One-way-ANOVA with post-hoc Tukey test was performed to determine statistical differences, if any, among the groups [61].

Ten days post infection the lesions measured approximately 5 mm in diameter. The infected mice were allocated to different groups (5 mice in each group) with similar mean lesion diameters in each group of mice (*p* > 0.5 by using one-way-ANOVA) and intravenously injected with 100 μL of blank or AmB-loaded chitosan nanoparticles intravenously (i.v.) over a period of 10 days, with formulation administration taking place on alternate days (5 doses in total, on days 0, 2, 4, 6 and 8). AmB dose was 5 mg/kg for chitosan-TPP particles and 10 mg/kg for chitosan-dextran particles (the different AmB doses reflect the safe AmB doses determined xii). Weight of nanoparticles in the blank formulations reflected the related AmB-loaded ones.

Another group of *Leishmania*-infected mice received 100 μL of the vehicle in the chitosan nanoparticles i.e., water intravenously (i.v.) for over 10 days, alternate day dosing (i.e., 5 doses in total, on days 0, 2, 4, 6 and 8).

A second in vivo experiment was conducted to confirm the initial findings and to establish a dose–response effect of AmB-loaded chitosan-TPP nanoparticles. Infected mice were intravenously administered 100 μL of the formulation at 5, 2.5 or 1.25 mg of AmB/kg for 10 days on alternate days on days 0, 2, 4, 6 and 8. The control group received blank chitosan-TPP nanoparticles, with nanoparticles’ weight being equivalent to that in animals receiving the 5 mg/kg dose.

For both experiments, the controls were as follows: (i) a negative control group where infected mice were left untreated; (ii) a positive control group where infected mice received 100 μL paromomycin (50 mg/kg) intraperitoneally for 10 consecutive days, a regimen with proven efficacy in this CL model [61,78] and (iii) a second positive control group which received 100 μL of AmBisome^®^ (size ≈ 80 nm) [24,25], at 10 mg of amphotericin B/kg intravenously (i.v.) on alternate days over 10 days, on days 0, 2, 4, 6 and 8.

Treatment efficacy was evaluated by measuring the daily the progression of lesion size, and the parasite load using bioluminescence and quantitative PCR (qPCR).

On day 10, mice were humanely killed and skin samples were harvested by surgical removal from the areas containing the localized CL lesion and non-CL-infected skin on the back (control site) of the same mouse and stored at −80 °C until the tissues were processed for measurement of parasite load by qPCR and measurement of AmB concentrations. The methodology used to extract parasite DNA from lesions and quantify parasite load by quantitative PCR was as described in detail by Wijnant et al. 2017 [78]. Briefly, qPCR methodology based on the amplification of the 170-bp region in the *Leishmania* 18 S gene was performed to quantify the parasite burden in the CL lesions using the RotorGene 3000 instrument, set at 40 cycles at a denaturation setting of 95 °C for 5 min followed by a 2-step amplification cycle of 95 °C for 10 s and 60 °C for 30 s. Qiagen DNeasy kit for blood and tissue was used to extract DNA from 200 µL volume of the homogenate. Extraction of AmB in CL lesion and non-lesion skin samples was performed as described in details by Wijnant et al. 2018 [22]. Briefly, the skin sample was homogenised in 3 cycles of 30 s at 6800 rpm using the Precellys homogeniser (Bertin Technologies, France) to obtain a smooth homogenate. Then, 100 µL of homogenate was added to a 250 µL of a mixture of methanol: DMSO (84:16) and 1 µL of 200 ng/mL tolbutamide (analytical standard; Sigma, UK) for drug extraction and protein precipitation. AmB was quantified by HPLC as described in Section 4.8, using a calibration curve prepared using AmB in untreated healthy skin homogenate.

Deoxycholate AmB was not included in our in vivo experiments as a control, as Wijnant et al. (2017) showed that the highest tolerated dose of deoxycholate AmB was 1 mg/kg administered intravenously in BALB/c mice, and after multiple administrations (1 mg/kg/iv on days 0, 2, 4, 6 and 8) no significant reduction in lesion sizes or parasite load was observed in in *L. major* infected mice [22].

Chitosan solution controls were not included for the (i) intravenous (iv.) administration due to viscosity (800–2000 cP at 1 wt.% in 1% acetic acid) [79,80]; (ii) intraperitoneal (i.p.) route as it was reported to cause extensive inflammatory reactions, leading to granulomas and adhesions in the peritoneal cavity in a rabbit model [81] and (iii) subcutaneous (s.c.) route as it promoted inflammatory reactions and a 67% cellular expansion in local lymph nodes in female C57 BL/6 mice model [82].

#### 4.12.2. Determination of Parasite Burden by Measuring Bioluminescence of the Leishmania Parasites

The luciferase substrate; luciferin (150 mg/kg, D-Luciferin potassium salt, Xenogen, Gold Biotechnology, St. Louis, MO, USA) was inoculated intraperitoneally (i.p.) into BALB/c mice 10 min before bioluminescent signals were acquired. After 7 min of injection, the mice were anaesthetized by inhalation with 3% isoflurane in oxygen (*v*/*v*) at a flow rate of 2.5 L/min until no movement was seen (3 min approx.). Mice were then imaged using the IVIS Spectrum in vivo Imaging System (PerkinElmer, Waltham, MA, USA), and the images were acquired with the Living Image software (version 4). Emitted photons were gathered by a charge couple device (CCD) camera (PerkinElmer, USA) using the medium resolution (medium binning) mode. A circular region of interest (ROI) encompassing the nodular area on the rump was drawn to quantify the bioluminescence, expressed as radiance in numbers of photons/sec [36].

### 4.13. Ex Vivo Investigation of AmB Permeation into CL Lesions Following Topical Application of Nanoparticles

Mice infected with *L. major* with CL lesions measuring approximately 5 mm in diameter (10 days following infection) were humanely killed and 2 circular discs (~15 mm diameter) of skin were excised from each mouse; one was from the CL lesion and the second was a non-infected piece of skin above the CL lesion on the higher back of the mouse. The skin discs were mounted between the donor and receptor compartments of Franz diffusion cells and held in place by a clamp. The receptor phase was PBS with 2% hydroxypropyl-β-cyclodextrin (CD); the latter was used to enhance AmB solubility (37 ug/mL AmB in PBS with 2% CD) in order to maintain sink conditions throughout the experiment [61]. A magnetic stirrer was introduced and the Franz cells were placed in a water bath at 34 °C on a magnetic stirrer plate set at a speed of 800 rpm [61]. Then, 100 µL of each formulation (AmB solution; AmB-CH-TPP; AmB-CH-Dex) containing 4 mg/mL AmB was applied to the skin in each donor compartment and the experiment was started. Subsequently, at regular time intervals, 100 µL of receptor fluid was taken and was replaced with 100 µL of fresh PBS with 2% CD. Samples of the collected receptor fluid were stored at −80 °C until analysed by HPLC. After 24 h the experiment was terminated, and the Franz cells were disassembled. Donor chambers were washed with 1 mL of methanol: DMSO (84:16) and the wash samples were stored at −80 °C until analysed by HPLC. A dry cotton swab was used to remove any residual AmB from the skin surface; the swab and the skin samples were stored at −80 °C until analysed by HPLC for AmB content. Extraction of AmB in the skin tissue after the permeation experiment, was performed as described in detail by Wijnant et al. 2018 [22]. Briefly, the skin sample was homogenised in 3 cycles of 30 s at 6800 rpm using the Precellys homogeniser (Bertin Technologies, France) to obtain a smooth homogenate. Then, 100 uL of homogenate was added to a 250 µL of a mixture of methanol: DMSO (84:16) and 1 µL of 200 ng/mL tolbutamide (analytical standard; Sigma, United Kingdom) for drug extraction and protein precipitation. AmB was quantified by HPLC as described in Section 4.8. The cumulative amount of drug that permeated through the skin as a function of time was plotted and the linear portion of the curve was used to calculate the flux and lag time (Figure 6).

The permeability coefficient was calculated according to the following equation:J = KpC_0_(5)
where J is the flux of the permeant per unit area (mol cm^−2^ s^−1^), Kp is the permeability coefficient and C_0_ is the concentration of drug applied to the skin surface (mol cm^−3^) [80].

## Figures and Tables

**Figure 1 molecules-25-04002-f001:**
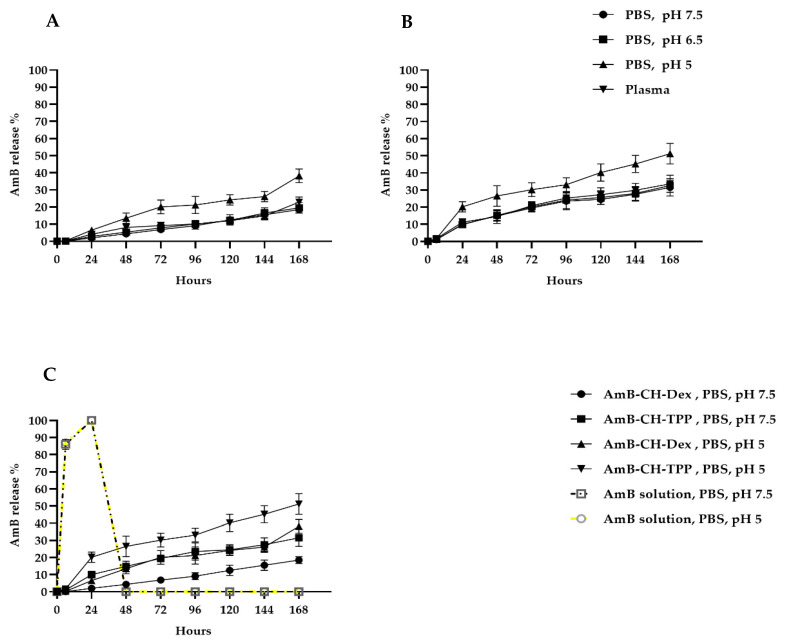
In vitro release profiles of AmB from chitosan nanoparticles at 37 °C. (**A**) AmB-CH-Dex in PBS (pH of 5, 6.5 or 7.5) and mouse (BALB/c) plasma, (**B**) AmB-CH-TPP in PBS (pH of 5, 6.5 or 7.5) and mouse (BALB/c) plasma. (**C**) A comparison of AmB release from AmB solution, AmB-CH-TPP and AmB-CH-Dex nanoparticles in PBS at pH 5 and 7.5. Data expressed as mean +/− SD (experiment was reproduced three times with confirmed similar data). AmB-CH-TPP size = 69 ± 8 nm and AmB-CH-Dex size = 174 ± 8 nm.

**Figure 2 molecules-25-04002-f002:**
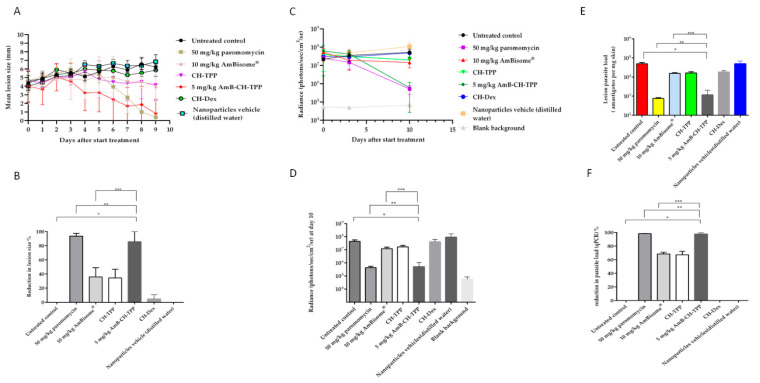
AmB nanoparticles efficacy in the lesion cure model in BALB/c mice infected with luciferase-expressing *L. major* parasites. *L. major* infected mice were allocated into 8 groups: (G1) represents untreated infected group, (G2) paromomycin as a positive control (50 mg/kg/QD for 10 days; i.p.), (G3) AmBisome^®^ as a comparison group (10 mg/kg/QAD for 10 days; i.v.), (G4) CH-TPP nanoparticles (mass of nanoparticles in the blank formulations reflected the related AmB-loaded ones) (QAD for 10 days; i.v.), (G5) AmB-CH-TPP nanoparticles (5 mg of AmB/kg/QAD for 10 days; i.v.), (G6) CH-Dex nanoparticles (mass of nanoparticles in the blank formulations reflected the related AmB-loaded ones) (QAD for 10 days; iv), (G7) AmB-CH-Dex nanoparticles (10 mg of AmB/kg/one dose; i.v.), 24 h after the first (and only) dose of the formulation, the mice looked unwell with piloerection and weight loss, therefore data of G7 are not represented in the figure or (G8) the nanoparticles vehicle (distilled water, QAD for 10 days; i.v.). QAD: every other day, QD: once a day. The average lesion size and parasite load represent the mean ± SD. One way-ANOVA for parasite load (bioluminescence signal), parasite load (qPCR) and repeated measures for lesion size followed by Tukey’s multiple-comparison tests was used to compare outcomes among the groups. A *p*-value < 0.05 was considered statistically significant ((*) *p* < 0.05, (**) *p* > 0.05 and (***) *p* < 0.05). (**A**) represents mean lesion size progression in function of time since the start of treatment, (**B**) represents the % reduction in lesion size compared with G1 (untreated infected group) at day 10, (**C**) represents the bioluminescence signal in function of time since the start of treatment, (**D**) represents the bioluminescence signal compared with G1 (untreated infected group) at day 10, (**E**) represents the parasite load (qPCR-DNA) at day 10 and (**F**) represents the % reduction in the parasite load (qPCR) compared with G1 (untreated infected group) at day 10.

**Figure 3 molecules-25-04002-f003:**
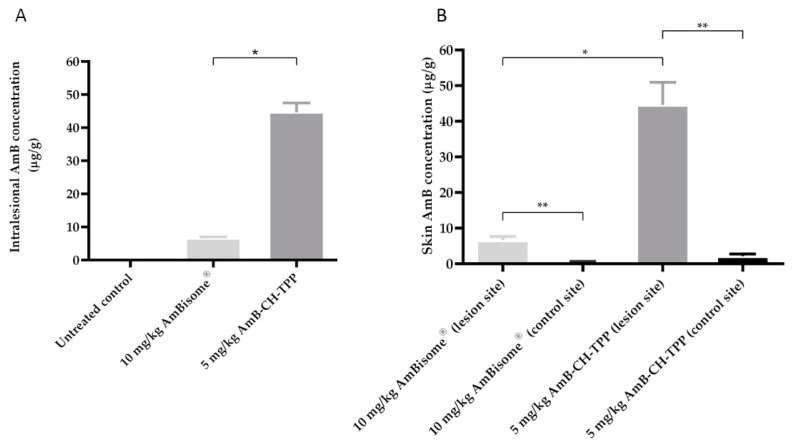
Multiple dose skin pharmacokinetics of AmB-CH-TPP nanoparticles and AmBisome. *L. major*-infected BALB/c mice received intravenous doses of AmBisome (G3, 10 mg/kg/QAD for 10 days; i.v.) and AmB-CH-TPP nanoparticles (G5, 5 mg of AmB/kg/QAD for 10 days; i.v.). 24 h after the last dosing, AmB levels in skin were determined. The CL lesion was localized on the rump, while the back skin of same mice was used as lesion-free, healthy control site. Each point represents the mean and standard error of the mean (*n* = 5 per group). (**A**) represents intralesional AmB and (**B**) represents a comparison between infected and uninfected skin AmB concentration. The data represent the mean ± standard error. ANOVA followed by Tukey’s multiple-comparison tests was used to compare outcomes among the groups. A *p*-value < 0.05 was considered statistically significant ((*) *p* < 0.05 and (**) *p* < 0.05).

**Figure 4 molecules-25-04002-f004:**
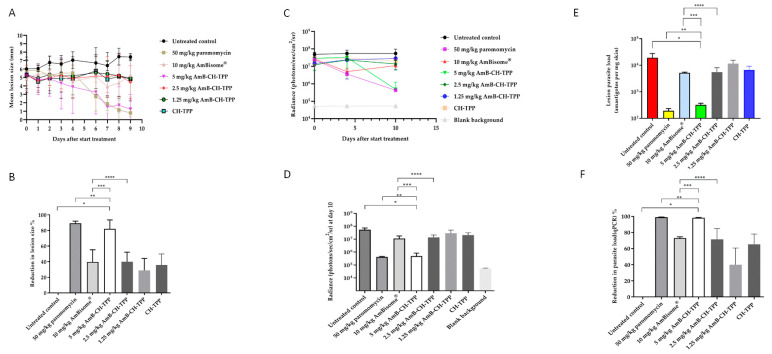
AmB nanoparticles efficacy in the lesion cure model in BALB/c mice infected with luciferase-expressing *L. major* parasites. *L. major* infected mice were allocated into 7 groups: (G1) represents untreated infected group, (G2) paromomycin as a positive control (50 mg/kg/QD for 10 days; i.p.), (G3) AmBisome^®^ as a comparison group (10 mg/kg/QAD for 10 days; i.v.), (G4) AmB-CH-TPP nanoparticles (5 mg of AmB/kg/QAD for 10 days; i.v.), (G5) AmB-CH-TPP nanoparticles (2.5 mg of AmB/kg/QAD for 10 days; i.v.), (G6) AmB-CH-TPP nanoparticles (1.25 mg of AmB/kg/QAD for 10 days; i.v.) and (G7) CH-TPP nanoparticles (nanoparticles mass being equivalent to that in animals receiving the 5 mg/kg dose) (QAD for 10 days; i.v.). The average lesion size and parasite load represent the mean ± SD. One-way-ANOVA for parasite load (bioluminescence signal), parasite load (qPCR) and repeated measures for lesion size followed by Tukey’s multiple-comparison tests was used to compare outcomes among the groups. A *p*-value < 0.05 was considered statistically significant ((*) *p* < 0.05, (**) *p* > 0.05, (***) *p* < 0.05 and (****) *p* > 0.05). (**A**) represents mean lesion size progression in function of time since the start of treatment, (**B**) represents the % reduction in lesion size compared with G1 (untreated infected group) at day 10, (**C**) represents the bioluminescence signal in function of time since the start of treatment, (**D**) represents the bioluminescence signal compared with G1 (untreated infected group) at day 10, (**E**) represents the parasite load (qPCR-DNA) at day 10 and (**F**) represents the % reduction in the parasite load (qPCR) compared with G1 (untreated infected group) at day 10.

**Figure 5 molecules-25-04002-f005:**
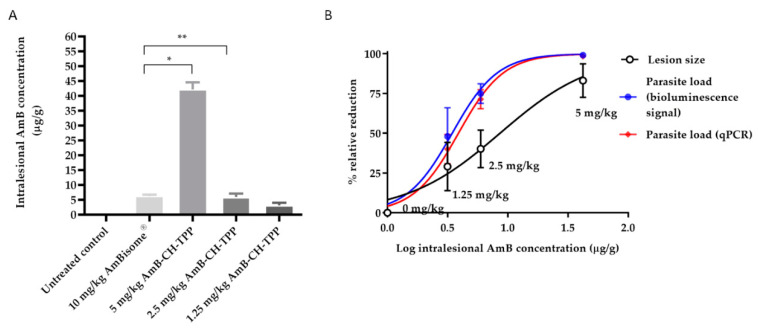
Multiple dose skin pharmacokinetics of AmB-CH-TPP nanoparticles and AmBisome^®^. *L. major*-infected BALB/c mice received intravenous doses of AmBisome^®^ (G3, 10 mg/kg/QAD for 10 days; i.v.), AmB-CH-TPP nanoparticles (G4, 5 mg of AmB/kg/QAD for 10 days; i.v.), AmB-CH-TPP nanoparticles (G5, 2.5 mg of AmB/kg/QAD for 10 days; i.v.) and AmB-CH-TPP nanoparticles (G6, 1.25 mg of AmB/kg/QAD for 10 days; i.v.). Then, 24 h after the last dosing, AmB levels in skin were determined. The CL lesion was localized on the rump, while the back skin of same mice used as lesion-free, healthy control site. Each point represents the mean and standard error of the mean (*n* = 5 per group). (**A**) represents intralesional AmB, (**B**) outcomes are linked in a logarithmic scale dose–response curve plotting drug concentrations against relative reduction in lesion size and parasite load measured using bioluminescence and qPCR. The data represent the mean ± standard error. ANOVA followed by Tukey’s multiple-comparison tests was used to compare outcomes among the groups. A *p*-value < 0.05 was considered statistically significant ((*) *p* < 0.05 and (**) *p* > 0.05).

**Figure 6 molecules-25-04002-f006:**
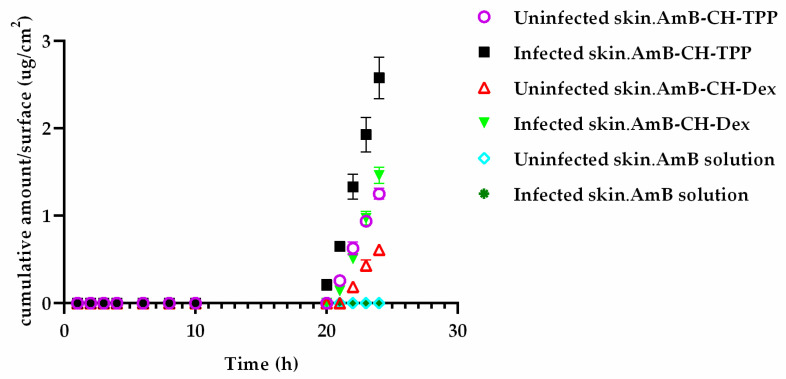
The cumulative amount of AmB permeated per surface area (μg/cm^2^) through uninfected BALB/c mouse skin (*n* = 5) and *L. major* infected BALB/c mouse skin (*n* = 5). Infected skin was more permeable to both types of AmB-loaded chitosan nanoparticles than uninfected skin (*p* < 0.05 by using repeated measures ANOVA). The use of AmB-CH-TPP nanoparticles enhanced AmB penetration through both healthy and infected skin compared to AmB-CH-Dex nanoparticles (*p* < 0.05 by using repeated measures ANOVA). AmB-CH-TPP nanoparticles (Size = 65 ± 8 nm, Zeta potential = 25.5 ± 1 mV) or AmB-CH-Dex nanoparticles (Size = 170 ± 8 nm, Zeta potential = −13 ± 1 mV).

**Table 1 molecules-25-04002-t001:** Physicochemical properties of blank and amphotericin B (AmB)-loaded chitosan nanoparticles.

		Nanoparticles			
		CH-TPP	AmB-CH-TPP	CH-Dex	AmB-CH-Dex
EE %	after lyophilization with sucrose 5%		94 ± 5		92 ± 8
AmB loading %	after lyophilization with sucrose 5%		26 ± 1		23 ± 2
yield %	after lyophilization with sucrose 5%	94 ± 4	93 ± 6	95 ± 3	92 ± 6
size (diameter) nm	before lyophilization	48 ± 6	57 ± 7	145 ± 6	164 ± 5
	after lyophilization with sucrose 5%	67 ± 7	69 ± 8	170 ± 9	174 ± 8
PDI	Before lyophilization	0.1 ± 0.01	0.1 ± 0.03	0.15 ± 0.01	0.16 ± 0.01
	after lyophilization with sucrose 5%	0.25 ± 0.05	0.2 ± 0.01	0.29 ± 0.04	0.26 ± 0.01
zeta potential mV	before lyophilization	32.1 ± 1.2	29 ± 2	−15.5 ± 1	−14 ± 2
	after lyophilization with sucrose 5%	28.5 ±1.9	25.5 ± 1	−12.9 ± 3	−11 ± 1

Data expressed as mean +/− SD (experiments were repeated three times and showed similar data). No significant difference was shown in % of EE, AmB loading and yield between AmB-CH-TPP and AmB-CH-Dex nanoparticles (*p* > 0.05 by using *t*-test). EE: encapsulation efficiency. PDI: polydispersity index.

**Table 2 molecules-25-04002-t002:** In vitro activity of chitosan formulations against intracellular *Leishmania* amastigotes at two pH values.

Compound	Properties	Medium pH = 7.5 *				Medium pH = 6.5 *			
		*L. major* **		*L. mexicana* **		*L. major* **		*L. mexicana* **	
		EC_50_ µg/mL	EC_90_ µg/mL	EC_50_ µg/mL	EC_90_ µg/mL	EC_50_ µg/mL	EC_90_ µg/mL	EC_50_ µg/mL	EC_90_ µg/mL
Amphotericin B (AmB solution)	Purity ≥ 95%, M_W_ 924.1	0.09 ± 0.003	0.5 ± 0.04	0.3 ± 0.003	0.7 ± 0.02	0.09 ± 0.003	0.5 ± 0.02	0.5 ± 0.04	0.6 ± 0.04
AmBisome^®^	Liposomal AmB, Size = 70–80 nm	1.2 ± 0.07	8 ± 0.3	1.8 ± 0.08	12 ± 1	1.3 ± 0.08	7 ± 0.1	1.8 ± 0.07	13 ± 1
HMW chitosan	M_W_ = 310–375 KDa	105 ± 7	1192 ± 58	123 ± 5	2206 ± 5	10 ± 0.3	127 ± 5	16 ± 0.7	165 ± 27
CH-TPP nanoparticles	Size = 67 ± 7 nm, Zeta potential = 28.5 ± 1.9 mV	162 ± 10	828 ± 43	177 ± 7	4020 ± 352	13 ± 0.5	122 ± 19	21 ± 0.9	284 ± 10
AmB-CH-TPP nanoparticles	Size = 69 ± 8 nm, Zeta potential = 25.5 ± 1 mV	0.14 ± 0.09	1 ± 0.09	0.5 ± 0.01	1.8 ± 0.1	0.06 ± 0.003	0.5 ± 0.08	0.3 ± 0.01	1.8 ± 0.02
CH-Dex nanoparticles	Size = 170 ± 9 nm, Zeta potential = −12.9 ± 3 mV	No activity up to 486							
AmB-CH-DEX nanoparticles	Size = 174 ± 8 nm, Zeta potential = −11 ± 1 mV	0.16 ± 0.008	1.4 ± 0.02	0.5 ± 0.01	1.8 ± 0.05	0.16 ± 0.007	0.9 ± 0.04	0.4 ± 0.01	1.8 ± 0.05
TPP	M_W_ = 367.864 g/mol	No activity up to 486							
dextran sulphate	M_W_ = 40 KDa	No activity up to 486							

Experiments were conducted in quadruplicate cultures, data expressed as mean +/− SD (experiment was reproduced further two times with confirmed similar data not shown). * Statistically significant differences were found for the EC_50_ values of chitosan or CH-TPP at pH = 6.5 and pH = 7.5 (*p* < 0.05 by using *t*-test). ** *L. major* amastigotes were significantly more susceptible to AmB solution and AmB-loaded chitosan nanoparticles than *L. mexicana* (*p* < 0.05 by using an extra sum-of-squares F test). AmB solution, AmB-CH-TPP and AmB-CH-Dex had a similar anti-leishmanial activity.

**Table 3 molecules-25-04002-t003:** Reduction lesion size and parasite load measured using bioluminescence and qPCR in treated groups compared to untreated group.

Compounds	% Reduction of Lesion Sizes	% Reduction of Parasite Load (Bioluminescence)	% Reduction of Parasite Load (qPCR)
paromomycin (50 mg/kg/QD for 10 days; i.p.)	89	99	99
AmBisome^®^ (10 mg/kg/QAD for 10 days; i.v.)	40	80	73
AmB-CH-TPP nanoparticles (5 mg of AmB/kg/QAD for 10 days; i.v.)	83	99	98
AmB-CH-TPP nanoparticles (2.5 mg of AmB/kg/QAD for 10 days; i.v.)	40	75	71
AmB-CH-TPP nanoparticles (1.25 mg of AmB/kg/QAD for 10 days; i.v.)	29	48	40
CH-TPP nanoparticles mass being equivalent to that in animals receiving the 5 mg/kg dose)	35	65	65

**Table 4 molecules-25-04002-t004:** A comparison between infected and uninfected skin AmB concentration in the lesion cure model in BALB/c mice infected with *L. major*.

AmB Concentration μg/g				
	AmBisome^®^ (10 mg/kg/QAD for 10 days)	AmB-CH-TPP (5 mg/kg/QAD for 10 days)	AmB-CH-TPP (2.5 mg/kg/QAD for 10 days)	AmB-CH-TPP (1.25 mg/kg/QAD for 10 days)
infected (lesion site)	6.3 ± 1	42 ± 5	5.9 ± 1	3.16 ± 0.8
uninfected (control site)	0.4 ± 0.1	1.9 ± 0.5	0.3 ± 0.1	0.12 ± 0.04

Data are expressed as mean ± SD.

**Table 5 molecules-25-04002-t005:** ED50 and ED90 of AmB-CH-TPP in BALB/c mice infected with *L. major*.

	Lesion Sizes	Parasite Load (Bioluminescence)	Parasite Load (qPCR)
ED_50_ mg/kg	2.5 ± 1.4	1.3 ± 0.7	1.5 ± 0.9
ED_90_ mg/kg	8.9 ± 4.1	3.9 ± 2	4 ± 2

Data are expressed as mean ± 95% confidence intervals [CI].

**Table 6 molecules-25-04002-t006:** Flux, lag time and the permeability coefficient (kp) for AmB-loaded chitosan nanoparticles.

Formulations	Flux (μg/cm^2^/h)		Lag Time (h)		Kp (cm/h)	
	Uninfected Skin	Infected Skin	Uninfected Skin	Infected Skin	Uninfected Skin	Infected Skin
AmB-CH-TPP nanoparticles	0.06 ± 0.002	0.12 ± 0.005	20 ± 0.1	19.8 ± 0.3	1.8 × 10^−5^ ± 0.05 × 10^−5^	3.15 × 10^−5^ ± 0.15 × 10^−5^
AmB-CH-Dex nanoparticles	0.04 ± 0.002	0.09 ± 0.002	20.5 ± 0.1	20.3 ± 0.02	0.9 × 10^−5^ ± 0.05 × 10^−5^	2.3 × 10^−5^ ± 0.06 × 10^−5^
AmB solution	0				0	

Data expressed as mean ± SD, *n* = 5. Statistically significant differences of flux and Kp were observed between uninfected and infected skin for both formulations (*p* < 0.05 by *t*-test). No statistically significant difference of lag time was observed between uninfected and infected skin for both formulations (*p* > 0.05 by using *t*-test). AmB-CH-TPP nanoparticles (Size = 65 ± 8 nm, Zeta potential = 25.5 ± 1 mV) or AmB-CH-Dex nanoparticles (Size = 170 ± 8 nm, Zeta potential = −13 ± 1 mV).

**Table 7 molecules-25-04002-t007:** Disposition of topically applied AmB from formulations following permeation experiment using healthy and *L. major* infected mouse skin.

Applied Formulations	After 24 h	% AmB Recovered of (Mean ± SD)		
		Uninfected Skin	*L. major* Infected Skin	*p*-Value *t*-Test Comparing Infected and Uninfected
AmB solution	on skin	94.65 ± 2	92.32 ± 1	>0.05
	in skin	5.35 ± 0.2	7.68 ± 0.2	>0.05
	through skin	0	0	>0.05
AmB-CH-TPP nanoparticles	on skin	69.92 ± 1	61.49 ± 1	<0.05
	in skin	29.85 ± 1	38.09 ± 0.5	<0.05
	through skin	0.23 ± 0.02	0.42 ± 0.05	<0.05
AmB-CH-Dex nanoparticles	on skin	81.65 ± 2	73.14 ± 2	<0.05
	in skin	18.23 ± 1	26.58 ± 1	<0.05
	through skin	0.12 ± 0.02	0.28 ± 0.02	<0.05

The amounts of AmB recovered from the different sites were expressed as a fraction of the applied amount. The average (± SD) percent for 5 infected mice is shown. *p*-values were determined by at test comparing uninfected and infected skin. AmB on skin detected in wash and cotton swab, AmB in skin extracted from skin homogenate and AmB through skin detected in receptor fluid. AmB-CH-TPP nanoparticles (Size = 65 ± 8 nm, Zeta potential = 25.5 ± 1 mV) or AmB-CH-Dex nanoparticles (Size = 170 ± 8 nm, Zeta potential = −13 ± 1 mV).

**Table 8 molecules-25-04002-t008:** HPLC parameters for AmB quantification.

HPLC Column	Injection Volume (uL)	Flow Rate (mL/min)	Mobile Phase	Detector Wavelength (nm)	Retention Time (min)
phenomenex; synergi–hydro RP (250 × 4.6 mm; 5 μm)	20	1	5 mM Na_2_EDTA in methanol	450	7.65

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
