# Peer review of "Activity of Amphotericin B-Loaded Chitosan Nanoparticles against Experimental Cutaneous Leishmaniasis"

_molecules, 2020, doi:10.3390/molecules25174002_

Round 1
Reviewer 1 Report
The current manuscript provides an account of chitosan based drug delivery for leishmaniasis intervention. Such systems are well known although with different preparation approaches. I recommend following revisions for the manuscript:
- The rationale of the research is not clear and appropriate for novelty. Making nanoparticles using different method is not novel - if the end products provides similar results. Emulsion and ionotopic methods have their own advantages and disadvantages. The drawbacks cited for emulsion approach and the notion that "we have used the ionotropic gelation method to prepare nanoparticles of a size smaller than 100nm" may not qualify for novelty. The rationale should have been based on performance and hence the introduction section needs revision.
- Figure 1: The SEM images need to be rescanned. Not clear and not representative.
- Figure 2: I do not understand the rationale of 10 days drug release for a topical drug delivery system - is it really applicable in case of leishmaniasis?
- Figure 3 onwards: Data related to AmB-CH-DEX is missing?
- Tables 3 and 4: The data between AmB-CH-TPP and AmB-CH-DEX is not very different - so the rationale of achieving less than 100nm for better outcome is doubtful?
- Figure 4: caption is missing?
- Number of references should be reduced.
Reviewer 2 Report
This paper by Riezk et al. reports the preparation, via ionic gelation, and study of two chitosan-based nanoparticles (NPs) as potential systems for topical delivery of amphothericin B (AmB) to tackle cutaneous leishmaniasis (CL).
Though the authors have carried out an extensive study, encompassing a considerable number of physico-chemical, in vitro, in vivo, and ex vivo assays, and included suitable controls and reference formulations, such as AmB in solution and the well-known Ambisome system, the work does not seem suitable for publication at this stage, on the following grounds: (i) it noticeably lacks novelty, (ii) it settles on a questionable scientific approach, as explained further below, and (iii) it fails to provide proof of concept on the working hypothesis originally formulated, which was the potential interest of the advanced formulations for topical treatment of CL.
Specifically, major concerns are:
(a) exploration of chitosan-based NPs for the delivery of AmB goes back for over a decade, and includes approaches that are quite similar to those herein reported, i.e., based on ionic pairing with dextran sulphate (see, e.g., ref. 36 of the manuscript that dates back from 2007) and TPP (see, e.g., ref. 38 of the manuscript that dates from 2015); it could be argued that most of past studies like these were focused on systemic, not topical, delivery, but precisely the present manuscript fails to offer interesting results on potential topical applications. The authors mention that they formulations perform better than Ambisome intravenously; but do they outperform other similar ones already published in the literature? If so, this should have been presented and thoroughly discussed, and focus of the report should instead fall onto systemic, not topical, applications; if not, one fails to identify where's the novelty in this report;
(b) the scientific approach is questionable, given that topical applications were the target of the reported work - if that was so, why waste animals for in vivo testing using the intravenous route, and why choose ex vivo assays to estimate topical efficacy, if this required excised skin from mice that were equally sacrificed? The reviewer couldn't find a clear reason for this. Why not use the animals directly for in vivo testing of topical efficacy (results might be surprising in spite of data obtained with the Franz diffusion cell...), and opt by ex vivo testing on human skin as a predictor of efficacy on human CL?
Minor issues are:
(c) if topical delivery was actually targeted, stability should have been tested in simulated wound fluid, not (only) in plasma;
(d) inconsistencies on significant figures are often found in the (too many) tables of the manuscript;
(e) lines 447 and 452 - the authors probably meant ex vivo, not in vitro, assays;
(f) line 526 - normality is an obsolete concentration unit; molarity should be used instead;
(g) inconsistencies in reference citations should be reviewed, including journal names (sometimes abbreviated, some others not) and typing problems as in, e.g., ref. 61.
(h) the paper has too many tables that make it too dense and not nice to read. Many of these tables could have been transferred into the Supplementary Information file; the most expressive example of this is Table 10 on page 39 (HPLC analysis conditions): it is totally unnecessary to include this in the manuscript.
Reviewer 3 Report
The paper is well organized and focused on an interesting field.
The paper can be accepted in the present form.
I think that it is very interesting as it focuses on the use of chitosan to obtain nanoparticles by means of the use of different cross linking agents. the paper is well written, all the methods are well presented and the description of the paper has been done in a ordered and well organized mode. Nanoparticles were properly characterized and in vitro studies underlined the effectiveness of them against leishmania microorganisms. The comparison with ambisome is appropriate and the results obtained after in vivo studies are properly presented. I do not have any concern for the publication of this paper in journal, only in table 1 i recommend to replace before lyophilizing with “Before liophilization”.Author Response
Please see the attachment.

Round 2
Reviewer 1 Report
I suggest the authors to include the response to figure 2 (in vitro study) in the text. No further comments.
